Employment status and sick-leave following obesity surgery: a five-year prospective cohort study

Andersen John Roger 1 2 johnra@hisf.no
Hernæs Ulrikke J.V. 2 3
Hufthammer Karl Ove 4
Våge Villy 2 5
1 Faculty of Health Studies, Sogn og Fjordane University College , Førde , Norway
2 Centre of Health Research, Førde Hospital Trust , Førde , Norway
3 Department of Research and Development, Haukeland University Hospital , Bergen , Norway
4 Centre for Clinical Research, Haukeland University Hospital , Bergen , Norway
5 Department of Surgery, Voss Hospital, Helse Bergen Health Trust , Voss , Norway
Nock Nora
Electronic publication date: 2015 Sep 29
Publication date: 2015
Volume: 3
Electronic Location ID: e1285
Received 2015 Mar 24; Accepted 2015 Sep 10
Copyright: © 2015 Andersen et al.
Copyright year: 2015
Copyright holder: Andersen et al.
License: This is an open access article distributed under the terms of the Creative Commons Attribution License, which permits unrestricted use, distribution, reproduction and adaptation in any medium and for any purpose provided that it is properly attributed. For attribution, the original author(s), title, publication source (PeerJ) and either DOI or URL of the article must be cited.
License URL: https://creativecommons.org/licenses/by/4.0/

Keywords: Work, Employment, Sick-leave, Predictors, Surgery, Obesity, Norway

Funding: Helse Vest RHF This study was funded by grant from Helse Vest RHF (the Western Norway Regional Health Authority). The funders had no role in study design, data collection and analysis, decision to publish, or preparation of the manuscript.

==============================
Background. Severe obesity is a risk factor for lower participation in paid work, but whether employment increases and sick leave decreases after obesity surgery is not well documented.

Methods. We assessed 224 Norwegian patients with severe obesity (mean age: 40; mean BMI: 49; 61% female) regarding employment status (working versus not working) and the number of days of sick leave during the preceding 12 months, before and five years after obesity surgery (75% follow-up rate). Logistic regression analysis was used to study preoperative predictors of employment status after surgery.

Results. There were no change in the employment rate over time (54% versus 58%), but the number of days of sick leave per year was significantly reduced, from a mean of 63 to a mean of 26, and from a median of 36 to a median of 4. Most of this change was attributable to patients with zero days of sick leave, which increased from 25% to 41%. Being female, older, having low education level, receiving disability pension and not being employed before obesity surgery were important risk factors for not being employed after obesity surgery. The type of obesity surgery, BMI and marital status were not useful predictors.

Conclusions. Our findings suggest that undergoing obesity surgery is not associated with a higher rate of employment, although it may reduce the number of days of sick leave. Additional interventions are likely needed to influence the employment status of these patients. The significant preoperative predictors of not being employed in this study provide suggestions for further research.

Introduction

Severe obesity, defined as having a body mass index (BMI) ≥ 40.0 or having obesity-related diseases and a BMI ≥ 35, has been associated with lower employment rates, largely because of the detrimental effect of obesity on health (Andersen et al., 2010; Gripeteg et al., 2012; Hawkins et al., 2007; Hernæs et al., 2014; Neovius et al., 2008). Studies have also shown that obese subjects are at increased risk for being discriminated against when applying for jobs, for being passed over for promotion and for being made redundant (Puhl & King, 2013). Thus, obesity has economic consequences both on an individual level and for families (Lund et al., 2011; Puhl & King, 2013). Consequently, increasing participation in paid work can be an important effect of the treatment of severe obesity. Such treatment can not only improve the well-being of individuals and their families, but also reduce the increasing indirect obesity-related financial costs faced by many societies (Lehnert et al., 2013).

Obesity surgery can be successful in terms of weight loss, the resolution of comorbidities and improvements in quality of life (Andersen et al., 2014; Colquitt et al., 2009). One hypothesis is that obesity surgery also leads to higher rates of employment; however, this is not well documented. Several studies (Andersen et al., 2010; Hawke et al., 1990; Hawkins et al., 2007; Martin et al., 1991; Narbro et al., 1999; Turchiano et al., 2014; Wagner, Fabry & Thirlby, 2007), but not all (Crisp et al., 1977; Gripeteg et al., 2012; Velcu et al., 2005), have suggested an overall positive effect of obesity surgery on employment status or sick leave. However, these studies were limited by small sample sizes (N < 80) (Andersen et al., 2010; Crisp et al., 1977; Hawkins et al., 2007; Van Gemert et al., 1999; Velcu et al., 2005; Wagner, Fabry & Thirlby, 2007), follow-up periods < 5 years (Andersen et al., 2010; Crisp et al., 1977; Hawke et al., 1990; Hawkins et al., 2007; Martin et al., 1991; Turchiano et al., 2014; Van Gemert et al., 1999; Wagner, Fabry & Thirlby, 2007) or by the use of outdated obesity surgery procedures such as ileojejunal bypass, non-adjustable gastric banding and vertical banded gastroplasty (Crisp et al., 1977; Gripeteg et al., 2012; Hawke et al., 1990; Narbro et al., 1999; Van Gemert et al., 1999). We also know little regarding preoperative predictors of employment status after obesity surgery. Providing additional information on these issues may be useful for further research on how to assist patients undergoing bariatric surgery to obtain and sustain participation in paid work.

In this paper we study employment status and sick leave before and five years after obesity surgery. We also study whether preoperative age, sex, marital status, education level, BMI, receipt of disability pension, employment status and type of obesity surgery predicted employment status five years after obesity surgery.

Material & Methods

Patients 18 years of age or older who were accepted for bariatric surgery at Førde Central Hospital in Norway between 2001 and 2008 were invited to participate in a prospective cohort study. Data were collected before and five years after surgery. The patients underwent biliopancreatic diversion with duodenal switch (BPD/DS), sleeve gastrectomy (SG), gastric bypass (GBP) or a conversion to BPDS/DS from gastric banding. During the first years of this study, BPD/DS was the primary choice of surgery at the hospital. This later changed to SG, as a part of a two-stage strategy, in which BPD/DS is regarded as a last-resort operation.

The study conforms to the principles outlined in of the Declaration of Helsinki and was approved by the Regional Committee for Medical and Health Research Ethics in Western Norway (REK vest, ref. nr. 2013/1747).

Assessments

Employment status and days of sick leave were assessed by self-report questionnaires. The patients were asked whether any of their income came from paid work (coded as yes or no) and to estimate their average percentage position in paid work (0–100%). They also reported the number of days with sick leave in the preceding 12 months. The validity of assessing this information by self-report has been shown to be good in the Norwegian general population (Myrtveit et al., 2013). Income that came from paid work at the time of the question (coded as yes or no) was further validated by correlating this variable with the actual income based on public data from the Norwegian Tax Administration (Spearman rank correlation = 0.87, p < 0.001) in a random subsample of the patients (n = 20).

Body weight was measured in light clothing without shoes, with a precision of 100 g. Height was measured in a standing position without shoes, with a precision of 1 cm. Weight and height were used to calculate the BMI (kg/m2). We also assessed the patients age, sex, marital status (married/cohabiting or not), education level (primary school, high school or university/college) and whether the patients received any disability pension at the time of the question (coded as yes or no).

Statistics

We performed the statistical analyses using IBM SPSS version 22.0 for Windows and R version 3.1.1 for Windows (R Core Team, 2014). All reported p-values are 2-sided, and p-values ≤ 0.05 are considered statistically significant. Continuous variables are reported as means, standard deviations, quartiles and/or 95% confidence intervals, whereas categorical variables are reported as counts and percentages. We used paired t-tests to test changes in continuous variables, and McNemar’s test to test changes in binary variables. To explore predictors of unemployment, we fitted logistic regression models with employment status after five years as the dependent variable. Explanatory factors were the preoperative variables age, sex, marital status, education level, BMI, receipt of a disability pension, employment status and type of surgery. Age and BMI were included in the analysis as continuous variables after testing for non-linearity. To detect any problems with multicollinearity in the predictors, we examined the generalised variance-inflation factors.

Results

By the five-year follow-up, we had employment data on 224 patients (75% follow-up rate) (Fig. 1 and Table 1). The overall rate of employment did not change over time, and was 54% at baseline and 58% at follow-up (p = 0.34; Table 2). Most individuals were either unemployed or worked full-time (83% at baseline and 82% at follow-up), and the mean full-time equivalent (i.e., the proportion of income that came from paid work) did not change over time (0.46 at baseline and 0.49 at follow-up; p = 0.54; Table 2). However, there were changes in employment status of the individual patients. Of the 102 patients who were not employed before surgery, 31 (30%) had become employed after five years, and of the 122 patients’ who were employed before surgery, 23 (19%) had lost their employment after five years (Fig. 3).

Figure 1 Study population flow chart.

Figure 2 Distribution of BMI before and five years after obesity surgery (density plots with jittered strip chart) (n = 224 at baseline, n = 219 at follow-up).

Table 1 Patient characteristics at baseline (n = 224).

	Mean/count	SD/(%)	
Age	40	9	
Sex			
Female	136	(61%)	
Male	88	(39%)	
Married/cohabitation	130	(58%)	
Education (n = 222)			
College/university	56	(25%)	
High school	107	(48%)	
Primary school	59	(27%)	
BMI	49	8	
Disability pension (n = 216)	70	(32%)	
Surgery method			
Biliopancreatric diversion with duodenal switch	154	(69%)	
Sleeve gastrectomy	51	(23%)	
Gastric bypass	5	(2%)	
Revisions	14	(6%)	

Table 2 Employment status and days per years with sick leave before and five years after obesity surgery (n = 224).

	Before operation	5 years after operation	P-value	
	Mean/count	SD/(%)	Quartiles	Mean/count	SD/(%)	Quartiles		
Employed (yes/no), count	122	(54%)	–	130	(58%)	–	0.34	
Full-time equivalent, meana	0.46	0.46		0.49	0.46		0.54	
Days with sick leave per yearb								
Patients employed at both baseline and follow-up (paired t-test, n = 75), mean	56	61	2; 40; 86.5	28	46	0; 5; 39	0.002	
Patients employed at at least one time point (n = 108 at baseline, n = 113 at follow-up), mean	63	73	1.5; 36; 108	26	45	0; 4; 35	–	
Notes.

a The fraction of full-time employment, e.g., 0 = unemployed, 0.5 = working half time, 1 = working full time.

b There was missing data on number of days with sick leave for some patients who stated they were employed (14 patients at baseline and 17 patients at follow-up). One patient reported being employed but having 365 days of sick leave. This was truncated to 260 days, the maximum possible number of working days.

Secondary stratified analyses showed that the employment rate in patients who received (some) disability pension before surgery (n = 70) was 17% at baseline and 21% at follow up (p = 0.55), while it was 71% at baseline and 74% at follow-up (p = 0.64) in patients who did not receive any disability pension before surgery (n = 146). We also found that the proportion of patients receiving disability pension did not change significantly (32% at baseline and 38% at follow up; p = 0.16).

Although the overall rate of employment remained unchanged, the number of days of sick leave per year was much reduced (Table 2). For patients who were employed at both time points, there was a reduction from a mean of 56 to a mean of 28 (p = 0.002) and from a median of 40 to a median of 5. Note that these estimates could be biased, as one might expect that the patients losing their job from baseline to follow-up (and thus not included in the above calculations) were patients with a large number of days of sick leave. We therefore also report the mean number of days at each time point (for all patients employed at each time point). The results are very similar, a reduction from 63 days to 26 days (means) or from 36 to 4 (medians). Most of this change was attributable to patients with zero days of sick leave, which increased from 25% (27/108) at baseline to 41% (46/113) at follow-up.

From the multiple logistic regression analysis we found that being female, being older, having a low education level (only primary school), receiving disability pension and/or not participating in paid work before surgery were important risk predictors for not being employed after obesity surgery. Marital status, BMI and type of obesity surgery were not useful as predictors (Table 3A). The predictor estimates did not change substantially when adjusted for other predictors, and the predictors showed good explanatory power (Tjur’s D = 0.45) (Tjur, 2009).

Table 3 Logistic regression for the risk of not being employed five years after obesity surgery.

	Unadjusted model	Adjusted model	
	ORa	95% CI	P-value	ORa	95% CI	P-value	
(A) For all patients (n = 211)	
Age (years)b	1.04	1.01 to 1.07	0.01	1.05	1.01 to 1.10	0.02	
Sex			<0.01			0.003	
Female (ref.)	1	– to –	–	1	– to –	–	
Male	0.34	0.18 to 0.61	<0.01	0.31	0.13 to 0.68	0.003	
Married/cohabitation	0.94	0.54 to 1.64	0.83	0.83	0.38 to 1.79	0.63	
Education			<0.001			<0.001	
University/college (ref.)	1	– to –	–	1	– to –	–	
High school	1.64	0.79 to 3.55	0.20	1.13	0.45 to 2.90	0.80	
Primary school	8.40	3.65 to 20.56	<0.001	6.98	2.41 to 21.73	<0.001	
BMI (kg/m2)b	1.01	0.97 to 1.04	0.74	1.03	0.98 to 1.08	0.30	
Disability pension before surgery	10.56	5.39 to 21.84	<0.001	4.05	1.68 to 10.07	0.002	
Not working before surgery	9.84	5.29 to 18.96	<0.001	6.40	2.85 to 15.05	<0.001	
Treatmentc			1.00			0.25	
Biliopancreatric diversion with duodenal switch	1	– to –	–	1	– to –	–	
Sleeve gastrectomy	0.98	0.51 to 1.89	0.96	1.59	0.63 to 4.11	0.33	
Revisions	0.98	0.31 to 2.97	0.98	0.38	0.06 to 1.94	0.26	
(B) For patients not receiving any disability pension before surgery (n = 144).	
Age (years)	1.01	0.97 to 1.06	0.49	1.05	1.00 to 1.10	0.05	
Sex			0.11			0.02	
Female (ref.)	1	– to –	–	1	– to –	–	
Male	0.54	0.25 to 1.15	0.11	0.33	0.12 to 0.82	0.02	
Married/cohabitation	0.91	0.43 to 1.94	0.81	0.96	0.38 to 2.46	0.94	
Education			0.001			0.001	
University/college (ref.)	1	– to –	–	1	– to –	–	
High school	1.20	0.46 to 3.41	0.72	0.90	0.29 to 2.89	0.85	
Primary school	5.39	1.88 to 16.84	0.002	5.47	1.66 to 19.92	0.007	
BMI (kg/m2)b	1.04	0.99 to 1.09	0.14	1.04	0.97 to 1.10	0.26	
Not working before surgery	4.24	1.94 to 9.48	<0.001	5.01	1.96 to 13.59	<0.001	
Treatmentc			0.70			0.52	
Biliopancreatric diversion with duodenal switch	1	– to –	–	1	– to –	–	
Sleeve gastrectomy	0.87	0.35 to 2.03	0.75	1.35	0.46 to 3.88	0.57	
Revisions	0.43	0.02 to 2.70	0.45	0.35	0.01 to 3.30	0.43	
Notes.

a OR > 1 means increased risk for not being employed in paid work five years after obesity surgery.

b Age and BMI were also included as non-linear terms (second-degree polynomials), with no notable changes in any estimated effects or p-values. We therefore only report the estimated linear effect.

c It was not possible to reliably estimate the effect of gastric bypass, as only 3 (out of 5) patients had complete follow-up data (all of them were employed at follow-up). The gastric bypass patients are therefore excluded from the models.

One could be concerned that the patients receiving disability pension has a very high risk of remaining unemployed, and therefore should be excluded from any analyses looking at predictors of being employed. As a sensitivity analysis, we therefore repeated the analysis but restricted to the patients not receiving any disability pension (Table 3B). The results are very similar to the ones based on complete data.

Of the 224 patients analysed above, 219 patients also had BMI data at follow-up. For these patients the mean BMI changed from 49.4 (SD: 8.0; CI [48.3–50.5]) at baseline to 31.3 (SD: 5.5; CI [30.6–32.0]) five years after surgery (p < 0.001) (Fig. 2). The change in BMI ranged from −0.5 to 40.5. The mean percent BMI loss was 35.8 (SD: 11.6).

Figure 3 Parallel set plot showing the number and percentage of patients employed before and five years after obesity surgery.

The widths of the lines are proportional to the number of patients.

Discussion

The rate of employment in this cohort was much lower both before (54%) and five years after obesity surgery (58%) than in the general Norwegian population (83%) with similar age and gender distribution (Andersen et al., 2010). Even though the employment rate did not increase after obesity surgery, the number of days of sick leave decreased significantly.

The previous literature on the effect of obesity surgery on employment status and sick leave shows mixed results. However, direct comparisons with our study are difficult, due to clinically and methodological differences (especially in the length of follow-up) and because the social context in other studies may have influenced work availability, access to social benefits and paid sick leave. Regarding long-term studies (≥5 years), we have only identified two studies other than ours that have reported participation or indicators of participation in paid work both before and after obesity surgery (Gripeteg et al., 2012; Velcu et al., 2005). The stable employment rate in the present study is comparable to findings of a US study (Velcu et al., 2005) that followed 41 patients who underwent GBP for five years, in which the rate of employment exhibited a statistically non-significant improvement from 34% to 44% (p = 0.13). Finally, in a Swedish study bariatric surgery was associated with a 17% (p = 0.01) reduction in disability pension for up to 19 years in men but not in women (Gripeteg et al., 2012).

The reduction in sick leave in the present study was large, and suggests that productivity was increased due to health benefits among those who had a paid job. We have not identified other long-term studies (≥5 years) on sick leave after obesity surgery. However, our findings are in agreement with a Swedish study reporting that patients aged 47–60 years who had undergone obesity surgery had 16% (p < 0.001) fever sick days than controls 2–3 years postoperatively (Narbro et al., 1999). However, no effect was found for patients younger than 47 years. In our study, the reduction in sick leave was not influenced by age (data not reported).

Our finding that preoperative status with respect to employment and disability pension predicted employment status after obesity surgery was as expected. Our study also suggests that being female, being older, having low education level (only primary school) and being unemployed or receiving disability benefits before surgery are important risk factors for not being employed after obesity surgery. Of these risk factors, only low education is modifiable. Thus, providing patients with education and targeted vocational rehabilitation might be a useful intervention.

The strengths of the present study are the long follow-up period and an acceptable attrition rate. Furthermore, the surgery procedures represent modern obesity surgery. However, the study also has certain limitations. First, we did not have a control group. Two previous observational studies examined unemployed patients with severe obesity by comparing outcomes in patients who underwent obesity surgery versus those who did not (Turchiano et al., 2014; Wagner, Fabry & Thirlby, 2007). Both studies found a significant improvement in employment rates in the surgical groups. However, we believe that this design may induce bias, as it does not include the possible risk that the obesity surgery is associated with a reduction in the rate of employment among those who were employed preoperatively. Thus, we believe that our naturalistic study provides more information on outcomes following obesity surgery, as it included all patients, regardless of preoperative employment status. It is possible that the rate of employment would have decreased significantly in a control group that was randomised to not having obesity surgery, especially if the alternative treatment had little effect on the patients’ health. To conduct a randomised controlled trial in this field is demanding, both practically and ethically (Sugerman & Kral, 2005). Because obesity surgery is currently the only known effective long-term treatment for severe obesity (Kwok et al., 2014), we likely have to rely on well-conducted prospective cohort studies (Wolfe & Belle, 2014).

One other limitation of our study is that our primary outcomes were based on self-reports, and recall bias may have occurred. However, we believe that the face validity regarding employment status at the time of the question is good, as it is quite easy to know whether one is employed in paid work. We also hypothesised that being employed was associated with higher actual overall incomes, and this was supported by the validation approach described in the methods section. On the other hand, we think that the information on the number of days of sick leave per year may have been influenced by recall bias. The recall bias could be systematic, for example in the form of an underestimation of the number of days of sick leave only after surgery. However, we believe that it is likely that the degree of recall bias was identical both before and after surgery. Thus, if the recall bias was unsystematic, our finding would remain valid.

Finally, we lacked information of the patient’s employment status in the years prior to seeking surgical treatment for their obesity. It is not unlikely that long-term preoperative unemployment is associated with lower chances of getting employed following obesity surgery. Thus, the inclusion of this information would increase the value of future studies.

Conclusions

In conclusion, the employment rate remained stable while the number of days of sick leave was reduced after obesity surgery. The reduction in days of sick leave is encouraging, and should be further studied in terms of replication of results and cost-effectiveness. The significant predictors of employment status in this study offer suggestions for future research. Providing patients with education and targeted vocational rehabilitation are potentially useful interventions. The stories of patients who joined or left the workforce after obesity surgery could be studied using qualitative methods. Finally, we recommend looking for novel additional interventions intended to increase the rate of employment in this patient group.

We thank Lisbeth Schelderup and the rest of the staff at the Obesity Clinic at Førde Central Hospital for conducting the data collection of this study.

Additional Information and Declarations

Competing Interests

Author Contributions

Human Ethics

Data Availability

The authors declare there are no competing interests.

John Roger Andersen conceived and designed the experiments, analyzed the data, contributed reagents/materials/analysis tools, wrote the paper, prepared figures and/or tables, reviewed drafts of the paper.

Ulrikke J.V. Hernæs conceived and designed the experiments, analyzed the data, wrote the paper, reviewed drafts of the paper.

Karl Ove Hufthammer analyzed the data, contributed reagents/materials/analysis tools, wrote the paper, prepared figures and/or tables, reviewed drafts of the paper.

Villy Våge conceived and designed the experiments, performed the experiments, analyzed the data, wrote the paper, reviewed drafts of the paper. Dr. Villy Våge started the obesity surgery program at Førde Central Hospital back in 2001, and set up a research data base that made this study possible. As a bariatric surgeon he also has operated on many of the patients included in this study.

The following information was supplied relating to ethical approvals (i.e., approving body and any reference numbers):

Regional Committee for Medical and Health Research Ethics in Western Norway (REK vest, ref. nr. 2013/1747).

The following information was supplied regarding data availability:

All relevant data can be dowladed here: 10.5281/zenodo.16272.

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
