# Peer review of "Employment status and sick-leave following obesity surgery: a five-year prospective cohort study"

_PeerJ, doi:10.7717/peerj.1285_

## Round 0.1 · original submission · Major Revisions

Please provide a point by point response detailing how and where in the revised manuscript each issue raised by each reviewer was addressed.

·

Basic reporting

Please see General Comments.

Experimental design

Please see General Comments.

Validity of the findings

Please see General Comments.

Additional comments

This study investigated employment status and days of sick leave in 224 obese men and women before and after obesity. There are several strengths to the study, including the inclusion of both genders, a comparatively large sample size (N), a follow-up period of 5 years, and a good follow-up rate of 75%. Most of my comments are minor, and they mostly pertain to what is missing in the paper. I believe these comments can satisfactorily be addressed by the authors with a revision. Please note that all of my comments are summarized below as "General Comments" and are grouped according to the sections in the manuscript.

Abstract
1. Line 29. Please change ‘not useful’ to ‘not significant’.

2. Line 32. Please change “these patients” to a more specific term…e.g., “individuals who undergo bariatric surgery for the treatment of obesity”, etc.

3. Line 33. I would consider deleting this final sentence—it is vague and the preceding 2 sentences seem to have already conveyed the main points of the study.

Introduction
1. The literature review could have been more specific. There are a lot of studies cited which are lumped together in a few sentences. The addition of a few more details would help better familiarize the reader with this literature and strengthen this section, for example, line 51-56 “these studies were limited by small sample sizes of approximately XX to XX (refs), short follow-up periods ranging on average from XX to XX months (refs), or by the use of outdated obesity surgery procedures such as XXXXX (refs).”

Methods
1. Line 77, 78. Please add a sentence of whether the definition of paid work includes both part-time and full-time work (“any income from paid work: yes/no”)? I am assuming that part-time and full-time are included, but this should be stated directly. It might also be mentioned again in the discussion that employment status in this study covered, without distinguishing between, part-time work and full-time work.

Results
1. Lines, 102-104.Please provide greater precision in the reporting of weight loss data to two decimal points “mean BMI changed from 49.XX (SD: 8.XX) to …… Please also add the range of mean BMI loss. Please also consider adding data on the mean weight loss in kg or % loss (IBW or EBW) in addition to BMI data.

2. In line with the above comment on the reporting of weight loss data, I have a conceptual point for the authors to consider: On average, this group is still obese at an average BMI of 31. The SD of 6 BMI units reveals variability in weight loss outcome, with some patients achieving a more successful weight loss at 5-years FUP than others. Perhaps not everyone had a successful surgery. The question is whether the amount of weight loss or the BMI at follow-up is related to employment status and days of sick leave at 5 years following surgery? Wouldn’t this be important to investigate and report, or to possibly control for? I believe the addition of this data into the analysis would considerably strengthen the paper and would improve the clinical, public health, and policy-making implications of this study.

3. Line 109. There are 23 patients who lost their employment during the 5 year FUP. Did the authors obtain data on reasons for becoming unemployed during the follow-up period? If yes, I would recommend adding these data to the paper. If not, I would suggest adding that reasons for becoming unemployed during the post-operative period are unknown, and this represents a potential limitation of the study’s conclusions regarding the specific or nuanced associations between bariatric surgery and employment. For example, to speculate, is it possible that surgical complications (or a poor weight loss outcome, see point #2 above) may have been a contributing factor to becoming newly unemployed during the post-operative period for some individuals?

4. Do you have data on the reduction in the proportion (%) of patients on disability? This would allow a direct comparison with the Gripeteg et al. (2012) study. At baseline, Table 1 shows that 70 (of 216) patients were receiving disability. How many were there at follow-up? This would be useful information and strengthen the paper.

Discussion
1. One of the main conclusions of the study is that only low education is modifiable. I would suggest that another finding ----that unemployment or receiving disability benefits prior to surgery significantly predicted unemployment following surgery---- is also worthy of emphasis. Based upon the main results of the study, targeted vocational rehabilitation efforts seem warranted for this high-risk group already at the pre-operative stage.

2. Line 183. I would also add that data on depression and other relevant comorbid medical and psychiatric comorbidity were lacking. These would have been useful and relevant factors in predicting unemployment status within this patient population who has documented high rates of comorbidity, and several of these factors may have significantly and independently contributed to unemployment.

·

Basic reporting

The Heading for table 3 is wrong.
L.165: "rate" is written twice.

Experimental design

The 5-year follow-up design is a clear strength and I agree with the authors that the naturalistic setting/cohort study is appropriate. The design and analytic procedure seem well justified with regards to the outcome "sick leave". However, I am strongly concerned about the design with regard to the outcome "employment". My basic concern is the inclusion of patients with disability pension in these analyses. It seems as if you use disability pension and employment as two different, independent, variables, which means that some unemployed patients are just unemployed. Other unemployed patients receive disability pension. The chance of employment is very different in these groups and it is unlikely that any patient with disability pension will ever be employed again and they should be excluded from these analyses. The interesting question would be whether unemployed patients who receive unemployment benefits have a greater chance of being employed after surgery. Please correct me if I have misunderstood your analyses, because this is potentially a major flaw.
On the other hand, if you could show that patients with disability pension give up their pension to work after surgery, this would be even more interesting.

Validity of the findings

The findings appear valid

Additional comments

Another concern is the use of very simple questions to assess employment and disability pension. It would really have strengthened the paper if we had information on whether it is full-time, part-time, temporary or permanent. Many of these patients may have increased their work participation, but we cannot see that. This needs more attention in the Discussion section. Along the same line sick leave is measured with self-reports which is slightly discussed. You are aware that recall bias may be a problem, but you do not provide any references to studies that have reported how difficult (or easy) it is to remember days of sick leave.

---

## Round 0.2 · accepted · Accept

All issues have been adequately addressed

·

Basic reporting

No further comments

Experimental design

No further comments

Validity of the findings

No further comments

Additional comments

The authors' response to point #11 was appreciated, but it was not included in the revision. It would have been appropriate to acknowledge this issue as a study limitation and possible future direction as part of the discussion section.

·

Basic reporting

No comments

Experimental design

No comments

Validity of the findings

No comments

Additional comments

The Authors have adressed my comments satisfactorily